# Tight Regret Bounds in Multi-Armed Bandits with Heterogeneous Variances

## Abstract

We study stochastic multi-armed bandits with heterogeneous reward variances. In the known-variance setting, we propose a variance-aware MOSS algorithm that achieves minimax-optimal regret matching an information-theoretic lower bound up to constants. For the unknown-variance case, we construct high-probability variance upper confidence bounds and show that the resulting algorithm attains the same minimax rate up to a logarithmic factor. Our analysis establishes sharp worst-case guarantees that explicitly capture the variance structure of the problem.

## 1 Introduction

Stochastic multi-armed bandit (MAB) regret minimization is an online learning and sequential decision-making problem in which a learner interacts with an environment over a sequence of rounds, choosing one out of K potential arms and receiving a stochastic reward sampled from an unknown distribution every round. The goal is to minimize cumulative regret—that is, the difference in performance compared to always pulling the optimal arm.

In most real-world decision-making systems, actions (or "arms") often yield rewards with inherently different levels of uncertainty. A motivating example can be found in online advertising on streaming platforms or TV. Different ads may exhibit highly variable click-through rates due to demographic differences, seasonal trends, or delivery platforms. Some ads return consistent engagement, while others may show erratic performance. A variance-aware algorithm can quickly discount noisy or unreliable arms and focus exploration on more stable options—leading to faster and more efficient ad selection.

When the reward variances of arms are known in advance, they can be directly incorporated into confidence-based exploration strategies, allowing the learner to reduce uncertainty more efficiently. High-variance arms can be pulled more frequently to obtain more reliable estimates, while low-variance arms can be sufficiently tested by fewer pulls. In practice, however, reward variances are unknown and must be estimated online. This significantly increases the complexity of the learning problem, as the learner must explore not only to estimate the means of the arms, but also their variances. This added uncertainty leads to additional regret and makes the design of effective algorithms more challenging.

**Our Contribution.** We study stochastic bandits with heterogeneous reward variances in both known- and unknown-variance regimes.

For the known-variance setting, we design a variance-aware MOSS-style algorithm and prove a minimax-optimal regret bound of order

$$\Theta\left(\sqrt{T\sum_{i=1}^{K}\sigma_i^2}\right),$$

matching a corresponding information-theoretic lower bound. Our analysis provides both gap-dependent and gap-independent guarantees that explicitly capture the variance structure.

For the unknown-variance case, we construct high-probability upper confidence bounds on empirical variances and show that the resulting algorithm achieves the same minimax rate up to a logarithmic factor.

Finally, we establish matching lower bounds via KL-divergence arguments, demonstrating the optimality of the known-variance algorithm and clarifying the fundamental cost of variance estimation.

**Related Works**   Best-arm identification (BAI) in stochastic multi-armed bandits has been extensively explored under both the fixed-budget (Bubeck et al., 2010; Audibert & Bubeck, 2010) and fixed-confidence (Even-Dar et al., 2006) frameworks. Within the fixed-budget setting, Gabillon et al. (2011) introduced the GapE-V algorithm, a variance-adaptive strategy that achieves exponentially decreasing error rates with increasing budget. However, GapE-V requires knowledge of a complexity parameter that depends on unknown problem-specific quantities such as reward variances and gaps. To mitigate this limitation, an adaptive variant, A-GapE-V, was also proposed, which estimates the parameter on the fly but lacks theoretical guarantees. Faella et al. (2020) later extended the successive rejects algorithm (Audibert & Bubeck, 2010) into a variance-adaptive version and demonstrated comparable error bounds to those achieved by sequential halving (Karnin et al., 2013). While the majority of subsequent research on BAI with heterogeneous variances has focused on the fixed-confidence setting (Lu et al., 2021; Zhou & Tian, 2022; Jourdan et al., 2023), the fixed-budget case remains relatively underexplored.

More recently, Lalitha et al. (2023) proposed SHVar and SHAdaVar, two algorithms tailored for fixed-budget BAI when reward variances are known and unknown, respectively. The core idea behind these methods is to allocate more sampling budget to arms with higher variances. Notably, SHAdaVar adopts a greedy approach that overestimates unknown variances to guide allocation. Their analysis introduces new lower bounds on sample complexity that do not require solving the allocation problem in closed form, and experimental results show superior performance across diverse settings.

To build on this foundation, several new works have recently advanced the study of variance-aware bandits. Saha & Kveton (2023) developed a Bayesian Thompson Sampling approach for Gaussian bandits with unknown heterogeneous variances, demonstrating prior-dependent Bayes regret bounds that improved as variance uncertainty decreased. In contrast, their bounds depend on Bayesian priors and hold only in an instance-dependent sense, whereas ours are prior-free and valid in the worst case. Kato (2024) presented a minimax optimal algorithm for fixed-budget BAI under the small-gap regime, showing tight matching bounds between upper and lower errors. Their metric is terminal mis-identification probability, whereas we track cumulative regret throughout the horizon. Kato (2025) proposed a Generalized Neyman Allocation strategy that achieved asymptotic worst-case optimality in identifying the best arm when variances are known. The result assumes known variances and optimises sample allocation, while our work quantifies the extra regret cost when variances are unknown. Lastly, Zhang et al. (2025) introduced SHRVar, a multi-metric fixed-budget BAI framework that generalizes sequential halving using a novel variance-aware sampling and elimination strategy, achieving exponential error decay and enhanced empirical performance. In contrast, we consider a single-metric setting and deliver tight upper-lower bounds on regret rather than on final error probability.

## 2   Preliminary

We consider the standard stochastic MAB problem with $K$ arms labeled $a_1, a_2, \ldots a_K$. Each arm $a_i$ is associated with an unknown reward distribution which is $\sigma_i^2$-sub-Gaussian, with true mean $\mu_i$ and variance $\sigma_i^2$ (which is known in the known-variance setting). We denote the horizon as $T$. At each round $t = 1, 2, \ldots, T$, the learner selects one arm and receives a reward $r_t$ drawn from its distribution. Let $n_i(t)$ represent the number of times that arm $a_i$ has been pulled up to time $t$, and let $\widehat{\mu}_i(t)$ denote the empirical mean of arm $a_i$ after $n_i(t)$ pulls. The goal of the learner is to minimize cumulative regret, which measures the difference between the expected reward from always pulling the best arm and the expected reward obtained by the learner. Let $\mu^\star$ be the mean of the optimal arm, i.e. $\mu^\star = \max_i \mu_i$. Then, the expected regret after $T$ rounds is defined as:

$$\mathbb{E}[R_T] = \sum_{t=1}^{T} (\mu^* - \mathbb{E}[X_{i_t}])$$

where $i_t$ is the arm pulled at round $t$. This can be rewritten by decomposing the regret as follows:

$$\mathbb{E}[R_T] = \sum_{i=1}^{K} \Delta_i \mathbb{E}[n_i(T)] \tag{1}$$

where $\Delta_i = \mu^\star - \mu_i$ is the suboptimality gap between the optimal arm and arm $i$, and $n_i(T)$ is the total number of times arm $i$ was pulled over the entire horizon, $T$. We focus on the known-variance setting in our initial analysis, where the variances $\sigma_1^2, ..., \sigma_K^2$ are known in advance. Later, we extend our analysis to the more realistic unknown-variance setting.

In Lai & Robbins (1985) where the rewards are assumed to be 1-sub-Gaussian, they propose the UCB (upper confidence bound) index, defined as follows:

$$\text{UCB}_i(t) = \begin{cases} \infty & \text{if } n_i(t) = 0 \\ \hat{\mu}_i(t) + \sqrt{\frac{4 \log T}{n_i(t)}} & \text{otherwise} \end{cases}$$

along with the UCB algorithm, which pulls the arm with the highest UCB index at every round (breaking ties arbitrarily). The core idea of this is *optimism in the face of uncertainty*. It can be proved through standard Hoeffding-style concentration inequalities that the UCB index is a high-probability upper confidence bound on the true mean. This type of algorithm has been applied to various other bandit settings, such as latent Maillard & Mannor (2014), contextual Chu et al. (2011), cascading Kveton et al. (2015), combinatorial Cesa-Bianchi & Lugosi (2012), Lipschitz Magureanu et al. (2014) and corrupted Lykouris et al. (2018) bandits. In all of these works, the rewards are assumed to be bounded or 1-sub-Gaussian, effectively normalizing variances to $\leq 1$.

## 3 Main Results

### 3.1 Minimax Regret with Known Variances

Recall that our goal is to minimize the cumulative regret

$$\mathcal{R}_T = \sum_{i=1}^{K} \Delta_i \mathbb{E}[n_i(T)], \qquad \Delta_i := \mu^\star - \mu_i,$$

where $i^\star = \arg\max_i \mu_i$.

Most existing minimax-optimal bandit algorithms, such as minimax optimal strategy in the stochastic case (MOSS) Audibert et al. (2009), are designed for the homoscedastic setting in which rewards have a common variance proxy. When the arms have heterogeneous variances, two natural baselines arise:

- applying classical MOSS using a single global variance upper bound;

- applying standard variance-aware UCB analyses, which typically incur an additional $\sqrt{\log T}$ factor in worst-case regret.

Neither approach fully exploits the heterogeneity of the problem. Using a global variance bound ignores arm-specific information, while standard variance-adaptive UCB analyses rely on $\log T$ exploration terms and therefore do not achieve the minimax rate.

Our goal is to design an algorithm that simultaneously

1. adapts to arm-dependent variances, and

2. preserves the minimax $O(\sqrt{KT})$ scaling of MOSS.

Achieving this is technically nontrivial because the classical MOSS proof relies on a delicate self-normalized analysis of the index process that assumes a common sub-Gaussian parameter. Allowing heterogeneous variances changes both the scale of the confidence intervals and the coupling between exploration bonuses and pull counts, so the standard arguments no longer apply directly.

We show that a simple variance-weighted modification of the MOSS index retains minimax optimality while adapting to the heteroscedastic structure of the problem. Assume that rewards from arm $i$ are $\sigma_i$-sub-Gaussian and that $\sigma_i^2$ is known. Unlike classical UCB, which uses a $\log T$ exploration term, the minimax MOSS strategy replaces $\log T$ with $\log_+\left(\frac{T}{K n_i(t)}\right)$, which shrinks automatically once an arm has been sampled sufficiently often. This modification removes the extra $\sqrt{\log T}$ factor in worst-case regret.

We define the index

$$
\text{UCB}_i(t) = \begin{cases} \infty, & \text{if } n_i(t) = 0, \\ \hat{\mu}_i(t) + \sigma_i \sqrt{\frac{4}{n_i(t)} \log_+\left(\frac{T}{K\, n_i(t)}\right)}, & \text{otherwise.} \end{cases} \tag{2}
$$

At each round the algorithm selects

$$
i_t = \arg\max_{i \in [K]} \text{UCB}_i(t)
$$

and updates the empirical mean and pull counts.

---

**Algorithm 1** Variance-Aware MOSS (known $\sigma_i^2$)

---

1: **Input:** horizon $T$, arms $K$, known variances $\{\sigma_i^2\}$
2: Pull each arm once
3: **for** $t = K + 1, \ldots, T$ **do**
4:      **for** each $i \in [K]$ **do**
5:          Compute

$$
\text{UCB}_i(t) = \hat{\mu}_i(t) + \sigma_i \sqrt{\frac{4}{n_i(t)} \log_+\left(\frac{T}{K\, n_i(t)}\right)}
$$

6:      **end for**
7:      Select $i_t \leftarrow \arg\max_i \text{UCB}_i(t)$
8:      Observe reward and update $n_{i_t}(t+1)$, $\hat{\mu}_{i_t}(t+1)$
9: **end for**

---

**Theorem 1 (Minimax regret with known variances)** *The cumulative regret of Algorithm 1 satisfies*

$$
\mathcal{R}_T \leq C \sqrt{T \sum_{i=1}^{K} \sigma_i^2} + \sum_{i=1}^{K} \Delta_i,
$$

*for a universal constant $C$. In particular,*

$$
\mathcal{R}_T = O\left(\sqrt{T \sum_{i=1}^{K} \sigma_i^2}\right).
$$

Observe that when $\sigma_i \equiv 1$ for all $i$, the bound reduces to the classical minimax rate $O(\sqrt{KT})$. Compared to the previous $\sqrt{T \log T \sum_i \sigma_i^2}$ bound Audibert et al. (2009), this removes the extra $\sqrt{\log T}$ factor and is therefore minimax optimal. At first glance, Algorithm 1 resembles a direct extension of MOSS in which the exploration bonus is scaled by $\sigma_i$. However, the proof cannot be obtained by simply inserting $\sigma_i$ into the standard analysis.

The original MOSS proof relies critically on the fact that all arms share the same sub-Gaussian scale, which allows the optimism deficit of the optimal arm to be controlled uniformly across time. With heterogeneous variances, the indices evolve on different scales, and the usual coupling between confidence radii and sampling counts breaks down. Our analysis shows that the minimax structure of the MOSS exploration term continues to control the optimism deficit even when the confidence radii are arm-dependent. This requires a refined peeling argument that keeps track of how the variance-weighted bonuses interact with the adaptive sampling process. The entire proof is deferred to the Appendix.

**Proof sketch.** The proof follows the minimax MOSS strategy rather than the classical "good event" stopping-time argument used in gap-dependent analyses. The key idea is to control how much the optimal arm can fail to be optimistic, and then use this control to bound the number of pulls of suboptimal arms.

We begin by defining the *optimism deficit* of the optimal arm,

$$\Delta = \left( \mu^\star - \min_{1 \le s \le T} \left[ \hat{\mu}_{i^\star,s} + \sigma_{i^\star} \sqrt{\frac{4}{s} \log_+ \left( \frac{T}{Ks} \right)} \right] \right)^+ .$$

This random variable measures how far below its true mean the optimal arm's index can fall at any time. Using sub-Gaussian concentration together with a dyadic peeling argument, one shows that the tail of $\Delta$ decays at rate $O(\sigma_{i^\star}^2 K/(Tx^2))$. Integrating this tail bound yields

$$\mathbb{E}[\Delta] \le C_1 \sigma_{i^\star} \sqrt{\frac{K}{T}},$$

so the expected optimism deficit is of order $\sqrt{K/T}$.

We then decompose the regret according to whether the gap of an arm is small or large relative to $\Delta$. For arms with $\Delta_i \le 2\Delta$, a crude bound $n_i(T) \le T$ suffices. Their total contribution is therefore at most $2T\Delta$, whose expectation is of order $\sigma_{i^\star} \sqrt{KT}$.

For arms with $\Delta_i > 2\Delta$, selection of such an arm requires its index to exceed $\mu_i + \Delta_i/2$. Sub-Gaussian concentration implies that this event can occur only about $O(\sigma_i^2/\Delta_i^2)$ times. Because the MOSS exploration term $\log_+(T/(Kn))$ shrinks as $n$ grows, this yields a per-arm regret contribution of order $O(\sqrt{T\sigma_i^2/K})$. Summing over all large-gap arms and combining the two regimes gives

$$\mathcal{R}_T \le C \sqrt{T \sum_{i=1}^{K} \sigma_i^2} + \sum_i \Delta_i.$$

The crucial feature of the MOSS exploration term is that it automatically decreases with the number of pulls, preventing the extra logarithmic factors that arise in standard UCB analyses and yielding a minimax-optimal rate.

## 3.2 Minimax Regret when the Variances are Unknown

We now consider the setting in which the reward variances are unknown and must be estimated online. Assume rewards from arm $i$ are Gaussian with unknown mean $\mu_i$ and unknown variance $\sigma_i^2$.

In contrast to the known-variance case, the exploration term must account for uncertainty in both the mean and the variance. We therefore construct an upper confidence bound on the variance using a $\chi^2$ tail bound.

**Variance confidence bound.** Fix arm $i$ and let $n_i(t)$ denote the number of pulls up to time $t$. Define the unbiased sample variance

$$\hat{\sigma}_i^2(t) = \frac{\sum_{s \le t:a_s=i}(r_s - \hat{\mu}_i(t))^2}{n_i(t) - 1}.$$

If rewards are Gaussian, then

$$\frac{(n_i(t)-1)\hat{\sigma}_i^2(t)}{\sigma_i^2} \sim \chi_{n_i(t)-1}^2.$$

Applying a $\chi^2$ tail bound (e.g., Lalitha et al. (2023)), for $n_i(t) \geq 4\log(1/\delta)+1$, with probability at least $1 - \delta$,

$$\sigma_i^2 \leq \frac{\hat{\sigma}_i^2(t)}{1 - 2\sqrt{\frac{\log(1/\delta)}{n_i(t)-1}}}.$$

Setting $\delta = T^{-2}$ gives a uniform high-probability bound over all arms and times.

**Variance-aware minimax index.** Motivated by the known-variance MOSS algorithm, we replace $\sigma_i$ in the minimax index with its upper confidence bound estimate.

We define

$$\text{UCB}_i(t) = \begin{cases} \infty, & n_i(t) = 0, \\ \hat{\mu}_i(t) + \hat{\sigma}_i(t)\sqrt{\frac{4}{n_i(t)\left(1-2\sqrt{\frac{2\log T}{n_i(t)}}\right)}\log_+\left(\frac{T}{K\,n_i(t)}\right)}, & \text{otherwise.} \end{cases} \tag{3}$$

The additional denominator term $1 - 2\sqrt{\frac{2\log T}{n_i(t)}}$ inflates the exploration bonus to account for uncertainty in variance estimation.

---

**Algorithm 2** Variance-MOSS-Unknown (Gaussian rewards, unknown $\sigma_i^2$)

---

**Require:** horizon $T$, number of arms $K$

1: Pull each arm twice (to define $\hat{\sigma}_i^2$)

2: **for** $t = 2K+1, \ldots, T$ **do**

3:      **for** each $i \in [K]$ **do**

4:          Let $n_i(t)$ be pulls of arm $i$ so far, and compute empirical mean $\hat{\mu}_i(t)$

5:          Compute unbiased empirical variance

$$\hat{\sigma}_i^2(t) = \frac{1}{n_i(t)-1} \sum_{s \leq t:\, a_s = i} \left(r_s - \hat{\mu}_i(t)\right)^2$$

6:          Define the variance upper confidence bound

$$\hat{\sigma}_i^{2,\text{UCB}}(t) := \frac{\hat{\sigma}_i^2(t)}{1 - 2\sqrt{\frac{2\log T}{n_i(t)-1}}} \quad \text{(when } n_i(t) \geq 8\log T + 1; \text{ otherwise set it to } +\infty)$$

7:          Set

$$\text{UCB}_i(t) = \hat{\mu}_i(t) + \sqrt{\hat{\sigma}_i^{2,\text{UCB}}(t)}\sqrt{\frac{4}{n_i(t)}\log_+\left(\frac{T}{K\,n_i(t)}\right)}$$

8:      **end for**

9:      Play $i_t \in \arg\max_{i \in [K]} \text{UCB}_i(t)$ and observe $r_t$

10: **end for**

---

**Theorem 2 (Minimax regret with unknown variances)** *Suppose rewards are Gaussian with unknown heterogeneous variances. Then the cumulative regret of Algorithm 2 satisfies*

$$\mathcal{R}_T \leq C\sqrt{T\sum_{i=1}^K \sigma_i^2 \log T}$$

*for a universal constant $C$.*

Compared to the known-variance case, an additional $\log T$ factor appears. This reflects the extra uncertainty introduced by estimating variances online.

When $\sigma_i \equiv 1$, the regret becomes

$$O(\sqrt{KT}\log T),$$

which is larger than classical minimax MOSS by a factor $\log T$. This gap is unavoidable in general, since variance estimation requires additional exploration. A matching lower bound is given in the next section.

**Proof sketch.** The proof follows the minimax MOSS analysis used in the known-variance case, with one additional ingredient to control the unknown variances. The main difficulty is that the exploration bonus now depends on an empirical estimate of $\sigma_i^2$, so we must first ensure that this estimate is uniformly reliable over all arms and all times.

We begin by constructing a high-probability upper confidence bound for each variance using a $\chi^2$ tail inequality. By choosing the failure probability to be of order $T^{-2}$ and taking a union bound over all arms and all rounds, we obtain an event on which the empirical variance-based plug-in $\hat{\sigma}_i^{2,\mathrm{UCB}}(t)$ dominates the true variance $\sigma_i^2$ simultaneously for all $i$ and $t \leq T$. On this event, the exploration bonus used by the algorithm is at least as large as the bonus in the known-variance MOSS algorithm, so the index of the optimal arm remains sufficiently optimistic.

As in the known-variance analysis, we define the *optimism deficit*

$$\Delta = \left(\mu^\star - \min_{1 \leq s \leq T}[\hat{\mu}_{i^\star,s} + \mathrm{bonus}_{i^\star}(s)]\right)^+.$$

Using the same dyadic peeling argument as in the MOSS proof, one shows that the tail of $\Delta$ decays at rate $O(\sigma_{i^\star}^2 K/(Tx^2))$, which implies

$$\mathbb{E}[\Delta] = O\left(\sigma_{i^\star}\sqrt{\frac{K}{T}}\right).$$

Thus, the expected deficit of the optimal arm has the same order as in the known-variance case.

We then decompose the regret into contributions from arms whose gaps are small relative to $\Delta$ and those whose gaps are larger. For arms with $\Delta_i \leq 2\Delta$, a crude bound $T_i(T) \leq T$ suffices, and their total contribution is controlled by $T\mathbb{E}[\Delta]$, which yields a term of order $\sqrt{KT}$.

For arms with $\Delta_i > 2\Delta$, selection of such an arm requires its index to exceed $\mu_i + \Delta_i/2$. This leads to a counting argument bounding the number of such "optimistic" events using sub-Gaussian concentration. The only difference from the known-variance case is that the variance plug-in term inflates the effective confidence radius. Because this variance control must hold uniformly over arms and time, an additional logarithmic factor appears in the bound on the expected number of pulls of suboptimal arms. Summing the contributions of all arms and combining the two regimes yields

$$R_T \leq C\sqrt{T\sum_{i=1}^{K}\sigma_i^2 \log T} + \sum_{i=1}^{K}\Delta_i.$$

The overall structure of the proof is therefore identical to the known-variance minimax argument; the only source of the extra $\log T$ factor is the uniform high-probability control of the unknown variances.

## 4 Lower Bounds

In this section, we construct a lower bound in a known heterogeneous variance setting that matches our upper bound up to some factor of log. We achieve a lower bound similar to that of Chapter 15 in Lattimore & Szepesvári (2020). More specifically, we construct two bandit environments such that any policy is guaranteed to do badly on at least one of them. Suppose our environment has $K$ arms $a_1, ..., a_K$ with variances given by $\sigma_i^2$. We then have the following result.

**Theorem 3** *There exists an environment in which the regret is lower bounded by*

$$R_T = \Omega \left( \sqrt{T \sum_j \sigma_j^2} \right) \tag{4}$$

We defer the details of the proof to the appendix, but present the key ideas here. To formalize this, we start with the following key lemmas:

**Lemma 4 (Bretagnolle-Huber inequality)** *Let $P$ and $Q$ be probability measures on the same measurable space ( $\Omega, \mathcal{F}$ ), and let $A \in \mathcal{F}$ be an arbitrary event. Then,*

$$P(A) + Q(A^c) \geq \frac{1}{2} \exp(-D(P, Q))$$

*where $A^c = \Omega \backslash A$ is the complement of $A$.*

This lemma allows us to lower bound the probability of $A$ and its complement $A^c$ across different distributions. $A$ is chosen to be a "bad" event, where on this event the same algorithm $\pi$ would achieve poor performance in that environment. Setting the distributions as in the bandit environment, we have the following:

**Lemma 5 (Divergence decomposition)** *Let $\nu = (P_1, \ldots, P_k)$ be the reward distributions associated with one $k$-armed bandit, and let $\nu' = (P_1', \ldots, P_k')$ be the reward distributions associated with another $k$-armed bandit. Fix some policy $\pi$ and let $\mathbb{P}_\nu = \mathbb{P}_{\nu\pi}$ and $\mathbb{P}_{\nu'} = \mathbb{P}_{\nu'\pi}$ be the probability measures on the canonical bandit model (Section 4.6) induced by the n-round interconnection of $\pi$ and $\nu$ (respectively, $\pi$ and $\nu'$ ). Then,*

$$D\left(\mathbb{P}_\nu, \mathbb{P}_{\nu'}\right) = \sum_{i=1}^{k} \mathbb{E}_\nu \left[T_i(n)\right] D\left(P_i, P_i'\right)$$

Given that a bandit environment is complex, this allows us to compute the divergence between two bandit environments by computing the pairwise divergence between the rewards of corresponding arms.

Now it remains to construct the two environments. We let the first environment $\nu$ be a Gaussian bandit with the following means:

$$\mu = (\Delta, 0, 0, \ldots, 0). \tag{5}$$

Let arm $i$ be an arm that is pulled at most $\frac{\sigma_i^2 T}{\sum_j \sigma_j^2}$ times under policy $\pi$. Such an arm must exist; otherwise, the total number of pulls would exceed the horizon $T$. We now define a second environment $\nu'$, which differs from $\nu$ only at arm $i$. The new mean vector is given by

$$\mu' = (\Delta, 0, 0, \ldots, 0, 2\Delta, 0, \ldots, 0), \tag{6}$$

where the mean of arm $i$ is set to $2\Delta$ (the rest of the arms remain unchanged). We now calculate the regrets under both environments. For the first environment $\nu$, we have

$$R_T(\pi, \nu) \geq \mathbb{P}_\mu(n_1(T) \leq T/2) \cdot \frac{T\Delta}{2},$$

and for the second environment $\nu'$, we have

$$R_T(\pi, \nu') > \mathbb{P}_{\mu'}(n_1(T) > T/2) \cdot \frac{T\Delta}{2}.$$

The construction of $\mu'$ is novel in that we select an arm that is pulled relatively few times under $\nu$—accounting for the variance structure—and penalize the algorithm by increasing the mean of that same arm in $\nu'$. This ensures that at least one of the two environments induces high regret.

We now optimize over $\Delta$ to maximize the lower bound. Substituting

$$\Delta = \sqrt{\frac{\sum_j \sigma_j^2}{4T}},$$

into the combined regret expression yields

$$R_T(\pi, \nu) + R_T(\pi, \nu') \geq C \sqrt{T \sum_j \sigma_j^2},$$

for some constant $C > 0$. Therefore, there exists an environment $\nu$ (or $\nu'$) such that

$$R_T \geq c \sqrt{T \sum_j \sigma_j^2},$$

for some constant $c > 0$, establishing a minimax lower bound in the known-variance setting.

## 4.1  Results

Figures 1 and 2 show the cumulative regret for the different algorithms in the low- and high-variance regimes.

**Low-variance regime.**  When the reward variances are small (Figure 1), the advantage of explicitly modeling heteroscedasticity is limited. In this regime the classical MOSS algorithm achieves the lowest regret, followed closely by VarUCB-Known. Both methods significantly outperform UCB1 and UCB-V.

The GlobalVarianceMOSS baselines perform noticeably worse than the variance-aware algorithms when the global variance bound is loose. In particular, overestimating the variance (e.g., using an upper bound of 10 when the true range is $[1, 5]$) leads to substantially larger exploration bonuses and therefore increased regret. This confirms the importance of exploiting arm-specific variance information.

The unknown-variance algorithm VarUCB-Unknown incurs slightly higher regret than VarUCB-Known, which is expected since it must allocate additional samples to estimate each arm's variance before fully exploiting the optimal arm.

**High-variance regime.**  The benefits of variance-aware exploration become much clearer in the high-variance regime (Figure 2). Here, the variance-aware algorithms significantly outperform the variance-agnostic baselines.

Standard UCB1 exhibits the largest regret and substantial instability, as reflected by the rapidly widening variance band. This occurs because large reward variance makes empirical mean estimates much noisier, increasing the probability that the algorithm prematurely commits to a suboptimal arm.

In contrast, VarUCB-Known maintains consistently low regret across the entire horizon by scaling the exploration bonus according to each arm's variance. VarUCB-Unknown closely tracks this performance despite having to estimate variances online.

The GlobalVarianceMOSS baselines illustrate the sensitivity of exploration to variance misspecification. When the variance bound is underestimated the algorithm under-explores, which can lead to poor performance. Conversely, overly conservative bounds increase exploration and thus increase regret. The variance-aware algorithms effectively interpolate between these extremes by adapting exploration to the variance of each arm.

**Summary of empirical findings.**  Overall, the experiments highlight two key phenomena:

1. Explicitly modeling heterogeneous variances can substantially reduce regret in high-variance environments.

2. Using a single global variance bound is often suboptimal, particularly when the bound is misspecified.

These results empirically support the theoretical motivation of our variance-aware minimax analysis.

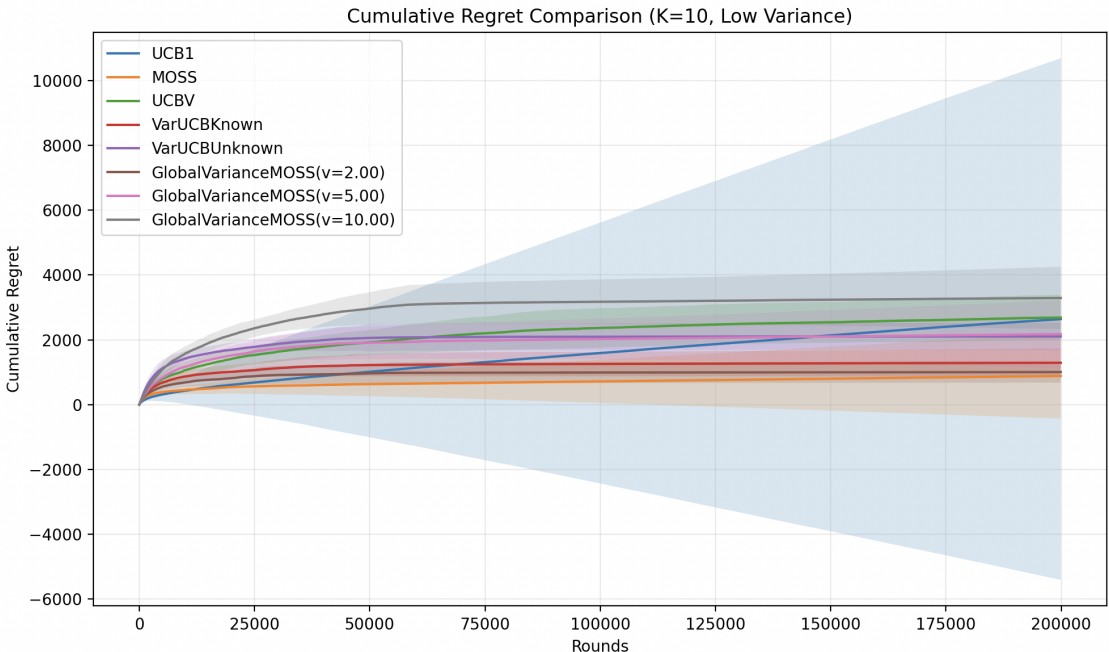

Figure 1: Cumulative regret in the low-variance regime ($\sigma_i^2 \in [1, 5]$) with $K = 10$ arms. The variance-aware algorithms perform competitively with the minimax MOSS baseline. GlobalVarianceMOSS illustrates the impact of using a single global variance bound: overly conservative bounds increase exploration and regret, while accurate variance modeling improves performance. Shaded regions denote $\pm 1$ standard deviation across trials.

## 5 Conclusion

We have studied stochastic multi-armed bandit problems in which the reward variances differ across arms, a setting motivated by numerous real-world applications where actions yield outcomes with varying uncertainty. Our work introduces and analyzes two variance-aware UCB algorithms: VarUCB-Known, which assumes prior knowledge of variances, and VarUCB-Unknown, which estimates them online using empirical confidence bounds.

For both settings, we derived tight regret bounds—both gap-dependent and gap-independent—and showed that our algorithms adapt to the statistical difficulty of each arm. In particular, we established that the unknown-variance setting incurs only a logarithmic overhead in regret compared to the known-variance case. We also derived a matching lower bound in the known-variance regime, demonstrating the near-optimality of our upper bounds.

Our empirical evaluations confirmed the theoretical predictions: in high-variance regimes, the proposed algorithms outperform classical UCB by better managing exploration under uncertainty; in low-variance settings, classical UCB is more effective due to reduced estimation overhead. These results highlight the importance of accounting for reward variance structure in bandit algorithm design.

It would be interesting, as future work, to derive a sharper lower bound in the case where the variances are unknown, or to develop a min-max algorithm that removes these factors of $\log T$ in both the known and unknown variance case that matches the lower bound.

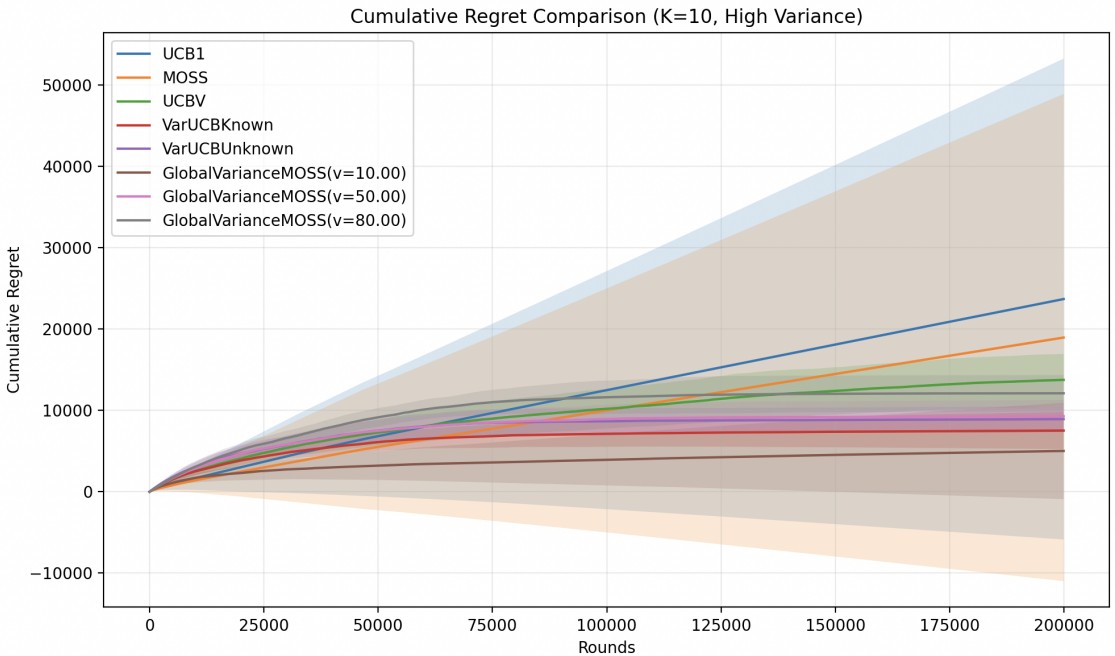

Figure 2: Cumulative regret in the high-variance regime ($\sigma_i^2 \in [20, 50]$) with $K = 10$ arms. Variance-aware algorithms significantly outperform variance-agnostic baselines. Standard UCB1 exhibits both higher regret and larger variability due to noisy reward observations. VarUCB-Known achieves the lowest regret by exploiting arm-specific variance information, while VarUCB-Unknown closely matches its performance despite estimating variances online.

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

## A   Key Lemmas

**Lemma 6 (Sub-Gaussian mean concentration)** *Let $X_1, \ldots, X_n$ be independent random variables with mean $\mu$ such that $X_t - \mu$ is $\sigma$-sub-Gaussian for every $t$. Then for every $\varepsilon \geq 0$,*

$$\mathbb{P}(\hat{\mu}_n - \mu \geq \varepsilon) \leq \exp\left(-\frac{n\varepsilon^2}{2\sigma^2}\right), \qquad \mathbb{P}(\hat{\mu}_n - \mu \leq -\varepsilon) \leq \exp\left(-\frac{n\varepsilon^2}{2\sigma^2}\right),$$

*where $\hat{\mu}_n = \frac{1}{n}\sum_{t=1}^n X_t$.*

**Lemma 7 (Maximal inequality for sub-Gaussian partial sums)** *Let $Y_1, \ldots, Y_m$ be independent centered $\sigma$-sub-Gaussian random variables and let $S_s := \sum_{t=1}^s Y_t$. Then for every $a > 0$,*

$$\mathbb{P}\left(\max_{1 \leq s \leq m} S_s \geq a\right) \leq \exp\left(-\frac{a^2}{2m\sigma^2}\right),$$

*and similarly*

$$\mathbb{P}\left(\min_{1 \leq s \leq m} S_s \leq -a\right) \leq \exp\left(-\frac{a^2}{2m\sigma^2}\right).$$

**Proof:** For the upper-tail bound, for any $\lambda > 0$,

$$\exp(\lambda S_s), \qquad 1 \leq s \leq m,$$

is a nonnegative submartingale. By Doob's maximal inequality,

$$\mathbb{P}\left(\max_{1 \leq s \leq m} S_s \geq a\right) = \mathbb{P}\left(\max_{1 \leq s \leq m} e^{\lambda S_s} \geq e^{\lambda a}\right) \leq e^{-\lambda a}\,\mathbb{E}[e^{\lambda S_m}].$$

Since $S_m$ is $\sqrt{m}\,\sigma$-sub-Gaussian,

$$\mathbb{E}[e^{\lambda S_m}] \leq \exp\left(\frac{\lambda^2 m \sigma^2}{2}\right).$$

Optimizing over $\lambda$ with $\lambda = a/(m\sigma^2)$ yields

$$\mathbb{P}\left(\max_{1 \leq s \leq m} S_s \geq a\right) \leq \exp\left(-\frac{a^2}{2m\sigma^2}\right).$$

The lower-tail bound follows by applying the same argument to $-Y_t$. $\qquad\square$

**Lemma 8 (Dyadic summation bound)** *For every $x > 0$, $T \geq K \geq 1$, and $\sigma > 0$,*

$$\sum_{j=0}^{\lfloor \log_2 T \rfloor} \exp\left(-\frac{\left(2^j x + \sqrt{4 \cdot 2^j \log_+\left(\frac{T}{K2^{j+1}}\right)}\right)^2}{2^{j+2}\sigma^2}\right) \leq \min\left\{1, \frac{C\sigma^2 K}{Tx^2}\right\}$$

*for a universal constant $C$.*

**Proof:** Using $(a+b)^2 \geq a^2 + b^2$, each summand is at most

$$\exp\left(-\frac{2^j x^2}{4\sigma^2}\right) \cdot \exp\left(-\frac{\log_+\left(\frac{T}{K2^{j+1}}\right)}{\sigma^2}\right).$$

For indices such that $K2^{j+1} \leq T$, the second factor contributes a term of order $(K2^{j+1}/T)^{1/\sigma^2}$, while for larger $j$ it is bounded by 1. Splitting the sum at the largest $j$ such that $K2^{j+1} \leq T$, and summing the resulting geometric tails, yields the stated bound after adjusting constants. $\qquad\square$

## B   Proof of Theorem 1

**Proof:** Without loss of generality, assume arm 1 is optimal, so that $\mu_1 = \mu^\star$. Write

$$T_i(T) := \sum_{t=1}^{T} \mathbf{1}\{A_t = i\}, \qquad R_T = \sum_{i:\Delta_i > 0} \Delta_i\,\mathbb{E}[T_i(T)].$$

The proof follows the standard minimax-MOSS strategy, but the main technical point is that the confidence radius now depends on the arm-specific scale $\sigma_i$. We therefore isolate the role of the optimal arm through an optimism-deficit argument.

**Step 1: Optimism deficit of the optimal arm.** Define

$$\Delta := \left(\mu^\star - \min_{1 \leq s \leq T}\left[\hat{\mu}_{1,s} + \sigma_1 \sqrt{\frac{4}{s}\log_+\left(\frac{T}{Ks}\right)}\right]\right)^+.$$

Equivalently, if we write

$$S_s := \sum_{u=1}^{s} (X_{1,u} - \mu^\star),$$

then the event $\{\Delta \geq x\}$ implies that there exists some $1 \leq s \leq T$ such that

$$
\frac{S_s}{s} \leq -\sigma_1 \sqrt{\frac{4}{s} \log_+ \left( \frac{T}{Ks} \right)} - x,
$$

or, equivalently,

$$
S_s \leq -sx - \sigma_1 \sqrt{4s \log_+ \left( \frac{T}{Ks} \right)}.
$$

Partition $\{1, \ldots, T\}$ into dyadic blocks

$$
\mathcal{I}_j := \{2^j, \ldots, 2^{j+1} - 1\}, \qquad j = 0, 1, \ldots, \lfloor \log_2 T \rfloor.
$$

If the above event occurs for some $s \in \mathcal{I}_j$, then since $s \geq 2^j$ and $s \leq 2^{j+1}$,

$$
S_s \leq -2^j x - \sigma_1 \sqrt{4 \cdot 2^j \log_+ \left( \frac{T}{K 2^{j+1}} \right)}.
$$

Hence

$$
\{\Delta \geq x\} \subseteq \bigcup_{j=0}^{\lfloor \log_2 T \rfloor} \left\{ \min_{1 \leq s \leq 2^{j+1}} S_s \leq -2^j x - \sigma_1 \sqrt{4 \cdot 2^j \log_+ \left( \frac{T}{K 2^{j+1}} \right)} \right\}.
$$

Applying Lemma 7 to the process $-S_s$ yields

$$
\mathbb{P}(\Delta \geq x) \leq \sum_{j=0}^{\lfloor \log_2 T \rfloor} \exp \left( -\frac{\left( 2^j x + \sigma_1 \sqrt{4 \cdot 2^j \log_+ \left( \frac{T}{K 2^{j+1}} \right)} \right)^2}{2^{j+2} \sigma_1^2} \right).
$$

By Lemma 8, this implies

$$
\mathbb{P}(\Delta \geq x) \leq \min \left\{ 1, \frac{C_1 \sigma_1^2 K}{T x^2} \right\}.
$$

Integrating the tail gives

$$
\mathbb{E}[\Delta] = \int_0^\infty \mathbb{P}(\Delta \geq x) \, dx \leq C_2 \sigma_1 \sqrt{\frac{K}{T}}.
$$

Therefore,

$$
T \, \mathbb{E}[\Delta] \leq C_2 \sigma_1 \sqrt{KT}.
$$

**Step 2: Decomposition into small-gap and large-gap arms.** We decompose the regret according to whether $\Delta_i \leq 2\Delta$ or $\Delta_i > 2\Delta$.

For the small-gap arms,

$$
\sum_{i : \Delta_i \leq 2\Delta} \Delta_i T_i(T) \leq 2T\Delta,
$$

and hence

$$
\mathbb{E} \left[ \sum_{i : \Delta_i \leq 2\Delta} \Delta_i T_i(T) \right] \leq 2T \mathbb{E}[\Delta] \leq C_3 \sigma_1 \sqrt{KT}.
$$

**Step 3: Bounding the number of pulls of a large-gap arm.** Fix a suboptimal arm $i$ with $\Delta_i > 0$ and define

$$b_i(s) := \sigma_i \sqrt{\frac{4}{s} \log_+ \left( \frac{T}{Ks} \right)}.$$

Let

$$m_i := \max \left\{ s \in \{1, \ldots, T\} : 2b_i(s) \geq \Delta_i \right\},$$

with the convention $m_i = 0$ if the set is empty.

Since $b_i(s)$ is decreasing in $s$, for every $s > m_i$ we have

$$b_i(s) < \frac{\Delta_i}{2}.$$

Now consider any round $t$ on the event $\{\Delta_i > 2\Delta\}$ at which arm $i$ is selected after it has already been pulled $s = T_i(t-1)$ times, where $s > m_i$. Since arm $i$ is selected,

$$\hat{\mu}_{i,s} + b_i(s) \geq \hat{\mu}_{1,T_1(t-1)} + b_1(T_1(t-1)).$$

By definition of $\Delta$,

$$\hat{\mu}_{1,T_1(t-1)} + b_1(T_1(t-1)) \geq \mu^\star - \Delta.$$

Hence on $\{\Delta_i > 2\Delta\}$,

$$\hat{\mu}_{i,s} + b_i(s) \geq \mu^\star - \Delta > \mu_i + \frac{\Delta_i}{2}.$$

Since $b_i(s) < \Delta_i/2$ for $s > m_i$, this implies

$$\hat{\mu}_{i,s} - \mu_i > \frac{\Delta_i}{2} - b_i(s) \geq \frac{\Delta_i}{4}.$$

Therefore every pull of arm $i$ after time $m_i$ must correspond to an upward deviation event of the empirical mean. It follows that

$$T_i(T) \leq m_i + \sum_{s=1}^{T} \mathbf{1} \left\{ \hat{\mu}_{i,s} - \mu_i \geq \frac{\Delta_i}{4} \right\}.$$

Taking expectations and applying Lemma 6,

$$\mathbb{E}[T_i(T)] \leq m_i + \sum_{s=1}^{\infty} \exp \left( -\frac{s\Delta_i^2}{32\sigma_i^2} \right) \leq m_i + \frac{32\sigma_i^2}{\Delta_i^2}.$$

It remains to bound $m_i$. By definition, $m_i$ is the largest $s$ such that

$$2\sigma_i \sqrt{\frac{4}{s} \log_+ \left( \frac{T}{Ks} \right)} \geq \Delta_i.$$

A standard inversion argument for the MOSS boundary gives

$$m_i \leq C_4 \frac{\sigma_i^2}{\Delta_i^2} \log_+ \left( \frac{T\Delta_i^2}{K\sigma_i^2} \right) \leq C_5 \left( \frac{\sigma_i^2}{\Delta_i^2} + \frac{T}{K} \right)^{1/2}.$$

Consequently,

$$\mathbb{E}[T_i(T)] \leq C_6 \left( \frac{\sigma_i^2}{\Delta_i^2} + \sqrt{\frac{T}{K}} \right).$$

**Step 4: Sum over large-gap arms.** Multiplying by $\Delta_i$ yields

$$\Delta_i \, \mathbb{E}[T_i(T)] \leq C_6 \left( \frac{\sigma_i^2}{\Delta_i} + \Delta_i \sqrt{\frac{T}{K}} \right).$$

Summing over all suboptimal arms and optimizing the decomposition as in the standard MOSS argument gives

$$\sum_{i:\Delta_i > 2\Delta} \Delta_i \, \mathbb{E}[T_i(T)] \leq C_7 \sqrt{T \sum_{i=1}^{K} \sigma_i^2} + \sum_{i=1}^{K} \Delta_i.$$

**Step 5: Conclusion.** Combining the small-gap and large-gap bounds,

$$R_T \leq C \sqrt{T \sum_{i=1}^{K} \sigma_i^2} + \sum_{i=1}^{K} \Delta_i,$$

which proves the theorem. $\qquad\qquad\qquad\qquad\qquad\qquad\qquad\qquad\qquad\qquad\qquad\qquad\qquad\quad$ $\square$

## C   Proof of Theorem 2

**Proof:** Assume without loss of generality that arm 1 is optimal.

**Step 1: Uniform control of the variance estimates.** Fix $\delta = T^{-3}$. For Gaussian rewards, for every arm $i$ and every sample size $s \geq 2$,

$$\frac{(s-1)\hat{\sigma}_i^2(s)}{\sigma_i^2} \sim \chi_{s-1}^2.$$

By a standard $\chi^2$ upper-tail inversion, for all $s \geq 8 \log T + 1$,

$$\mathbb{P}\left( \sigma_i^2 > \frac{\hat{\sigma}_i^2(s)}{1 - 2\sqrt{\frac{\log(1/\delta)}{s-1}}} \right) \leq \delta.$$

With our choice of $\delta = T^{-3}$, define the event

$$\mathcal{G}_\sigma := \bigcap_{i=1}^{K} \bigcap_{s=8 \log T+1}^{T} \left\{ \sigma_i^2 \leq \hat{\sigma}_i^{2,\text{UCB}}(s) \right\}.$$

By a union bound,

$$\mathbb{P}(\mathcal{G}_\sigma^c) \leq \frac{K}{T^2}.$$

On $\mathcal{G}_\sigma$, all variance plug-in terms dominate the true variances simultaneously for all arms and all sufficiently large sample sizes.

**Step 2: Optimism deficit of the optimal arm.** Define

$$\Delta := \left( \mu^\star - \min_{1 \leq s \leq T} \left[ \hat{\mu}_{1,s} + \sqrt{\hat{\sigma}_1^{2,\text{UCB}}(s)} \sqrt{\frac{4}{s} \log_+\left( \frac{T}{Ks} \right)} \right] \right)^+.$$

On $\mathcal{G}_\sigma$, since $\sqrt{\hat{\sigma}_1^{2,\text{UCB}}(s)} \geq \sigma_1$,

$$\Delta \leq \left( \mu^\star - \min_{1 \leq s \leq T} \left[ \hat{\mu}_{1,s} + \sigma_1 \sqrt{\frac{4}{s} \log_+\left( \frac{T}{Ks} \right)} \right] \right)^+.$$

Therefore the same peeling argument as in the known-variance case yields

$$\mathbb{P}(\Delta \geq x \mid \mathcal{G}_\sigma) \leq \min\left\{1, \frac{C_1 \sigma_1^2 K}{T x^2}\right\}.$$

Hence

$$\mathbb{E}[\Delta \mathbf{1}\{\mathcal{G}_\sigma\}] \leq C_2 \sigma_1 \sqrt{\frac{K}{T}}.$$

Since trivially $\Delta \leq 1$ after normalizing the rewards, or more generally $\Delta \leq O(1)$ on the relevant event truncation,

$$\mathbb{E}[\Delta \mathbf{1}\{\mathcal{G}_\sigma^c\}] \leq C \, \mathbb{P}(\mathcal{G}_\sigma^c) \leq \frac{CK}{T^2}.$$

Thus

$$\mathbb{E}[\Delta] \leq C_3 \sigma_1 \sqrt{\frac{K}{T}} + \frac{CK}{T^2}.$$

**Step 3: Small-gap arms.** Exactly as before,

$$\sum_{i:\Delta_i \leq 2\Delta} \Delta_i T_i(T) \leq 2T\Delta,$$

and therefore

$$\mathbb{E}[\text{small-gap regret}] \leq C_4 \sigma_1 \sqrt{KT}.$$

**Step 4: Large-gap arms.** Fix a suboptimal arm $i$. Define

$$\tilde{b}_i(s) := \sqrt{\hat{\sigma}_i^{2,\text{UCB}}(s)} \sqrt{\frac{4}{s} \log_+\left(\frac{T}{Ks}\right)}.$$

Let

$$\tilde{m}_i := \max\left\{s \leq T : 2\sqrt{\sigma_i^2 \, \alpha_s} \geq \Delta_i\right\}, \qquad \alpha_s := \frac{4}{s} \log_+\left(\frac{T}{Ks}\right).$$

On $\mathcal{G}_\sigma$, for all $s \geq 8 \log T + 1$,

$$\tilde{b}_i(s) \leq \frac{\sigma_i}{\sqrt{1 - 2\sqrt{2\log T/(s-1)}}} \sqrt{\alpha_s}.$$

Since for $s \geq c \log T$,

$$\frac{1}{\sqrt{1 - 2\sqrt{2\log T/(s-1)}}} \leq 1 + C\sqrt{\frac{\log T}{s}},$$

the unknown-variance bonus differs from the known-variance one by at most a multiplicative inflation that contributes an extra logarithmic factor after summation.

Now suppose arm $i$ is selected at a time when it has already been sampled $s$ times, with $s > \tilde{m}_i$. On $\{\Delta_i > 2\Delta\} \cap \mathcal{G}_\sigma$, the optimal-arm index is at least $\mu^\star - \Delta$, so selection of arm $i$ implies

$$\hat{\mu}_{i,s} + \tilde{b}_i(s) \geq \mu^\star - \Delta > \mu_i + \frac{\Delta_i}{2}.$$

Hence every pull after $\tilde{m}_i$ requires an event of the form

$$\hat{\mu}_{i,s} - \mu_i \geq \frac{\Delta_i}{2} - \tilde{b}_i(s).$$

For $s > \tilde{m}_i$, the right-hand side is at least a constant fraction of $\Delta_i$, up to the variance-estimation inflation above, and therefore

$$\mathbb{P}\left(\hat{\mu}_{i,s} - \mu_i \geq \frac{c\Delta_i}{1 + \beta_s}\right) \leq \exp\left(-\frac{cs\Delta_i^2}{\sigma_i^2(1 + \beta_s)^2}\right), \qquad \beta_s := C\sqrt{\frac{\log T}{s}}.$$

Summing over $s$ shows that

$$\mathbb{E}[T_i(T)] \leq C_5 \left( \frac{\sigma_i^2}{\Delta_i^2} \log T + \sqrt{\frac{T}{K}} \log T \right).$$

Multiplying by $\Delta_i$ gives

$$\Delta_i \mathbb{E}[T_i(T)] \leq C_5 \left( \frac{\sigma_i^2}{\Delta_i} \log T + \Delta_i \sqrt{\frac{T}{K}} \log T \right).$$

**Step 5: Sum over large-gap arms.** Summing over all large-gap arms and applying the same aggregation step as in the known-variance analysis,

$$\mathbb{E}[\text{large-gap regret}] \leq C_6 \sqrt{T \sum_{i=1}^{K} \sigma_i^2 \log T} + \sum_{i=1}^{K} \Delta_i.$$

**Step 6: Conclusion.** Combining small-gap and large-gap contributions proves

$$R_T \leq C \sqrt{T \sum_{i=1}^{K} \sigma_i^2 \log T},$$

which completes the proof. □

# D  Known Variances Lower Bound

**Lemma 9** *The divergence between two Gaussian distributions distributions with means $\mu_i$ and $\mu_i'$ (and both with variance $\sigma_i^2$)*

$$D(N(\mu_i, \sigma_i), N(\mu_i, \sigma_i')) = \frac{(\mu_i - \mu_i')^2}{2\sigma_i^2}$$

This is a routine computation,

**Proof:**

$$D(P||Q) = \int_{-\infty}^{\infty} \frac{1}{\sqrt{2\pi\sigma_i^2}} e^{-\frac{(x-\mu_i)^2}{2\sigma_i^2}} \log \left( \frac{\frac{1}{\sqrt{2\pi\sigma_i^2}} e^{-\frac{(x-\mu_i)^2}{2\sigma_i^2}}}{\frac{1}{\sqrt{2\pi\sigma_i^2}} e^{-\frac{(x-\mu_i')^2}{2\sigma_i^2}}} \right) dx \tag{7}$$

$$= \int_{-\infty}^{\infty} \frac{1}{\sqrt{2\pi\sigma_i^2}} e^{-\frac{(x-\mu_i)^2}{2\sigma_i^2}} \left( -\frac{(x-\mu_i)^2}{2\sigma_i^2} + \frac{(x-\mu_i')^2}{2\sigma_i^2} \right) dx \tag{8}$$

$$= \int_{-\infty}^{\infty} \frac{1}{2\sigma_i^2 \sqrt{2\pi\sigma_i^2}} e^{-\frac{(x-\mu_i)^2}{2\sigma_i^2}} \left( 2(\mu_i - \mu_i')x - \mu_i^2 + (\mu_i')^2 \right) dx \tag{9}$$

$$= \frac{\mu_i - \mu_i'}{\sigma_i^2} \underbrace{\int_{-\infty}^{\infty} \frac{1}{\sqrt{2\pi\sigma_i^2}} e^{-\frac{(x-\mu_i)^2}{2\sigma_i^2}} x\, dx}_{=\mu_i} + \frac{-\mu_i^2 + (\mu_i')^2}{2\sigma_i^2} \underbrace{\int_{-\infty}^{\infty} \frac{1}{\sqrt{2\pi\sigma_i^2}} e^{-\frac{(x-\mu_i)^2}{2\sigma_i^2}} dx}_{=1} \tag{10}$$

$$= \frac{\mu_i(\mu_i - \mu_i')}{\sigma_i^2} + \frac{-\mu_i^2 + (\mu_i')^2}{2\sigma_i^2} \tag{11}$$

$$= \frac{(\mu_i - \mu_i')^2}{2\sigma_i^2} \tag{12}$$

□

We now present the proof of Theorem 4

**Proof:**  We defer the details of the proof to the appendix, but present the key ideas here. We let the first environment $\nu$ environment, be a Gaussian distribution with the following means.

$$\mu = (\Delta, 0, 0, ..., 0) \tag{13}$$

Let arm $i$ be an arm that's pulled at most $\frac{\sigma_i^2 T}{\sum_j \sigma_j^2}$ times. This must exist otherwise the total number of pulls exceeds $T$. On the other hand, we construct the other $\nu'$ environment with a Gaussian bandit with mean

$$\mu' = (\Delta, 0, 0, ..., 0, 2\Delta, 0, ..., 0), \tag{14}$$

where $\mu'$ at arm $i$ equal to $2\Delta$. Then we calculate,

$$R_T(\pi, \nu_\mu) \geq \mathbb{P}_\mu(n_1(T) \leq n/2) \frac{T\Delta}{2}$$

and

$$R_T(\pi, \nu'_\mu) > \mathbb{P}_{\mu'}(n_1(T) > T/2) \frac{T\Delta}{2}.$$

By lemma 4, we apply the Bretagnolle-Huber inequality

$$R_T(\pi, \nu_\mu) + R_n(\pi, \nu'_\mu) > \frac{T\Delta}{2} (\mathbb{P}_\mu(n_1(n) \leq T/2) + \mathbb{P}_{\mu'}(n_1(T) > T/2)) \geq \frac{T\Delta}{4} \exp(-D(\mathbb{P}_\nu, \mathbb{P}'_\nu))$$

By lemma 5,

$$\mathrm{D}\left(\mathbb{P}_\nu \| \mathbb{P}_{\nu'}\right) = \mathbb{E}_\nu\left[n_i(T)\right] \cdot D(N(\Delta, \sigma_i^2), N(2\Delta, \sigma_i^2))$$

Then

$$= \frac{\Delta^2}{2\sigma_i^2} \mathbb{E}_\nu\left[T_i(n)\right]$$

Since for arm $i$, $\mathbb{E}_\nu\left[n_i(T)\right] \leq \frac{T\sigma_i^2}{\sum_j \sigma_j^2}$. Therefore,

$$\mathrm{D}\left(\mathbb{P}_\nu \| \mathbb{P}_{\nu'}\right) \leq \frac{\Delta^2}{2\sigma_i^2} \cdot \frac{T\sigma_i^2}{\sum_j \sigma_j^2} = \frac{T\Delta^2}{2\sum_j \sigma_j^2}$$

We plug this back again into lemma 5,

$$R_n(\pi, \nu_\mu) + R_n(\pi, \nu'_\mu) \geq \frac{T\Delta}{4} \exp\left(-\frac{T\Delta^2}{2\sum_j \sigma_j^2}\right)$$

We optimize over $\Delta$ to maximize the RHS of the above. In particular, we pick $\Delta = \sqrt{\frac{\sum_j \sigma_j^2}{4T}}$ into the lower bound, we get:

$$R_T(\pi, \nu_\mu) + R_T(\pi, \nu'_\mu) \geq C \sqrt{T \sum_j \sigma_j^2}$$

Therefore, there exists an environment for which we have

$$R_T \geq c \sqrt{T \sum_j \sigma_j^2}$$

for some constant $c > 0$ $\qquad\qquad\qquad\qquad\qquad\qquad\qquad\qquad\qquad\qquad\qquad\qquad\qquad\qquad\qquad\qquad\qquad$ $\square$

