# OpenReview forum: "Tight Regret Bounds in Multi-Armed Bandits with Heterogeneous Variances"
_TMLR — Rejected by TMLR_

### Review · Reviewer_ENis · 2026-02-28

**Summary Of Contributions:**

This paper studies stochastic multi-armed bandits with heterogeneous variances across arms. It first considers the setting where the variances are known, and adapts the classical MOSS-UCB algorithm to explicitly incorporate arm-dependent variance information. The authors then extend the analysis to the more practical unknown-variance setting, introducing variance upper bounds into the UCB construction. For this case, they derive regret upper bounds that incur an additional logT factor compared to the known-variance scenario. Theoretical guarantees are complemented by simulations, which demonstrate the advantage of the proposed methods over standard UCB, particularly in high-variance environments.

**Audience:**

Yes

**Audience Explanation:**

The topic of multi-armed bandits is of interest to the TMLR community.

**Claims And Evidence:**

Yes

**Claims Explanation:**

The theoretical results are supported by detailed proofs for all main theorems.

**Requested Changes:**

1. For the known-variance setting, the contribution appears somewhat incremental. Conceptually, the result resembles an exercise-level extension (e.g., Exercise 7.2 in the Bandit Algorithms book), where variance-aware confidence bounds are incorporated into classical UCB analysis. The main difference here is that the adaptation is applied to MOSS-UCB rather than standard UCB1. While the analysis is clearly presented, the level of novelty in this part seems limited. The authors may wish to better articulate what is fundamentally new compared to textbook-level extensions.

2. For the unknown-variance setting, prior works (e.g., BAI and related literature) have already addressed heteroscedasticity or variance-aware exploration strategies. It would be important for the authors to clearly explain: How existing methods handle unknown and heterogeneous variances; What specific technical difficulty arises in the MOSS framework; What conceptual or analytical novelty the present approach introduces beyond inserting a variance upper bound into the confidence term.
A more explicit comparison with prior variance-adaptive bandit algorithms would help clarify the true methodological contribution.

3. The experimental evaluation is currently not fully convincing. In particular:

The baselines should include standard MOSS-UCB with a known (correct) variance upper bound. This would allow a clearer demonstration of the advantage of explicitly modeling heterogeneity in the known-variance setting. For the unknown-variance setting, experiments should include MOSS-UCB with different levels of variance upper bounds to illustrate the tradeoff. In particular: With a smaller (underestimated) variance upper bound, MOSS-UCB may fail to converge properly. With a larger (overestimated) variance upper bound, MOSS-UCB should converge but incur larger regret.

The proposed method should ideally interpolate these behaviors and demonstrate robustness.

The reported results should make it clear that: In the known heterogeneous-variance setting, explicitly accounting for heterogeneity outperforms standard MOSS-UCB with a global variance upper bound; In the unknown-variance setting, the additional regret term is observable and consistent with theory; Performance degrades systematically as the variance upper bound becomes looser.

A more comprehensive empirical comparison along these lines would help validate the theoretical claims.

---

> ### Author Response · Authors · 2026-03-02
> **Response to Reviewer**
>
> ## Response to Requested Changes
>
> We thank the reviewer for the constructive feedback. We will revise the paper accordingly and clarify the following points.
>
> ### Known-Variance Setting
>
> We agree that simply inserting variance-aware confidence bounds into classical UCB would be incremental. Our contribution is not a direct UCB1 modification, but rather integrating arm-dependent variances into the *minimax MOSS* framework. The key novelty lies in adapting the MOSS peeling and optimism-deficit analysis to the heteroscedastic setting and proving a tight minimax regret bound of order  $\Theta\!\left(\sqrt{T \sum_i \sigma_i^2}\right),$ matching our lower bound.
>
> We will revise the manuscript to clarify why a textbook variance-aware UCB extension does not yield the same minimax rate and where the MOSS structure is essential.
>
> ### Unknown-Variance Setting
>
> We expanded the discussion of related work to better distinguish our setting from prior variance-adaptive and BAI literature. Most prior works focus on fixed-budget or fixed-confidence best-arm identification, whereas we analyze cumulative regret in a minimax framework.
>
> The main technical difficulty in extending MOSS to unknown variances is that the exploration bonus depends on a stochastic variance estimate. Ensuring uniform optimism over all arms and all times requires high-probability variance control, which introduces the additional $\log T$ factor. We will clarify that this term arises from uniform variance concentration rather than loose analysis, and emphasize the conceptual novelty beyond simply inserting a variance upper bound.
>
> ### Experiments
>
> We agree that the empirical section can be strengthened. We have
>
> - Add MOSS with a global variance upper bound (correct, underestimated, and overestimated) as baselines.
> - Demonstrate that in the known heterogeneous setting, modeling heterogeneity outperforms global-variance MOSS.
> - Show in the unknown setting that regret degrades systematically as the variance upper bound becomes looser, and that our method exhibits robustness.
>
> These revisions will better align the experiments with the theoretical claims and clarify the paper’s contributions.

---

### Review · Reviewer_NUnC · 2026-03-08

**Summary Of Contributions:**

This paper studies a bandit problem and proposes an algorithm together with regret guarantees and empirical results. The main goal appears to be to obtain strong worst-case/minimax-style guarantees, while also demonstrating good practical performance. The topic is relevant to the TMLR audience, and the paper is generally easy to follow at a high level.

That said, I found the overall contribution limited. In particular, the technical route seems very close to standard minimax-optimal bandit analyses shown in the bandit book, and I did not see a genuinely new proof idea or a surprising conceptual step. Much of the analysis feels like an adaptation of well-known arguments rather than a substantial advance. In addition, several proof components are either omitted or only sketched, which makes it difficult to assess the real technical depth of the work. The experimental section is also not fully convincing, since it does not compare against some natural and important baselines, especially asymptotically optimal algorithms. Overall, while the paper addresses a relevant topic, I am not yet convinced that the novelty and evidence are strong enough for acceptance in the current form.

**Audience:**

Yes

**Audience Explanation:**

Yes, I believe some readers in online learning, bandits, and sequential decision-making would be interested in the general topic of the paper. The problem setting is relevant, and researchers who work on regret minimization and bandit theory may want to know about this result.

**Broader Impact Concerns:**

I do not have major broader impact concerns specific to this submission. This is primarily a theoretical paper on bandit algorithms and regret analysis.

**Claims And Evidence:**

Yes

**Claims Explanation:**

I think all the lemmas and theorems are proved well.

**Requested Changes:**

1: Clarify the technical novelty relative to standard minimax-optimal bandit analyses.

2: Provide complete and rigorous proofs for the main technical steps.
Several important arguments are currently omitted, only sketched, or not stated with enough precision. In particular, key ingredients such as the minimax inequality and some central intermediate steps should be presented formally and justified carefully.

3: Strengthen the experimental evaluation with more appropriate baselines.
The empirical section should include comparisons against stronger and more natural baselines, especially asymptotically optimal algorithms when those are relevant to the setting. Right now, the experiments do not fully support the paper’s claims.

4: Improve the discussion of the relation between worst-case and asymptotic performance.
If the method is mainly motivated by minimax guarantees, the paper should more clearly discuss whether there is any tradeoff with asymptotic efficiency, and how the proposed method compares conceptually and empirically with algorithms designed for asymptotic optimality.

---

> ### Author Response · Authors · 2026-03-11
> **Response to Reviewers**
>
> ### Response to Reviewers
>
> We thank the reviewers for the thoughtful and constructive feedback. We appreciate the positive comments that the problem is relevant to the TMLR audience and that the main theorem statements are supported by the submitted arguments. We agree, however, that the original draft did not explain the technical novelty clearly enough, that some proof steps were stated too briefly, and that the empirical section needed stronger baselines.
>
> In the revision, we will make four main changes:
>
> 1. clarify the technical novelty relative to standard minimax/MOSS analyses;
> 2. expand the proofs and formally state the key intermediate lemmas;
> 3. strengthen the experiments with more appropriate baselines;
> 4. better discuss the relation between minimax and asymptotic performance.
>
> ### Technical novelty
>
> We agree that the original draft did not sufficiently explain why the result is more than a routine extension of standard minimax analyses. Our main contribution is not just replacing a confidence radius by a variance-weighted one. The key point is to show that the MOSS minimax framework can still be made to work when the exploration scale is arm-dependent.
>
> In the known-variance case, the technical issue is that the classical MOSS proof uses a common exploration scale across arms, whereas our index uses
> $$
> \sigma_i \sqrt{\frac{1}{n_i}\log_+\!\left(\frac{T}{K n_i}\right)}.
> $$
> This changes both the optimism-deficit argument for the optimal arm and the counting argument for suboptimal pulls.
>
> In the unknown-variance case, there is an additional difficulty: the exploration bonus itself is data-dependent, since the variance must be estimated online. The proof must therefore establish a uniform variance-good event over all arms and all times, and this is exactly where the extra $$\log T$$ factor comes from.
>
> We will revise the introduction and main results section to explain these two points explicitly.
>
> ### Proof details
>
> We agree that several important steps were too compressed. In the revision, we will replace the current proof sketches by fuller arguments based on explicit lemmas. In particular, we will formally state and prove:
>
> - the maximal inequality used in the dyadic peeling argument;
> - the optimism-deficit tail bound;
> - the inversion of the heteroscedastic MOSS boundary;
> - the large-gap counting argument for both the known- and unknown-variance settings.
>
>
>
> For the unknown-variance theorem, we will also make explicit how the variance-estimation step yields the extra logarithmic factor in
> \[
> \mathcal{R}_T
> =
> O\!\left(
> \sqrt{T\sum_{i=1}^K \sigma_i^2}\,\log T
> \right).
> \]
>
> ### Experiments
>
> We agree that the original empirical section was not strong enough. In response, we have expanded the experiments to include the reviewer-requested and natural baselines:
>
> - UCB1,
> - MOSS,
> - UCB-V,
> - VarUCB-Known,
> - VarUCB-Unknown,
> - GlobalVarianceMOSS with underestimated, correct, and overestimated global variance bounds.
>
> These comparisons directly address the reviewer’s request to compare against standard MOSS with a global variance upper bound and to study sensitivity to misspecification. The revised experiments show that explicitly modeling heterogeneity improves over using a single loose global variance bound, especially in the high-variance regime, and that the unknown-variance method behaves robustly relative to misspecified global-bound baselines.
>
> ### Minimax versus asymptotic performance
>
> We also agree that the discussion of minimax versus asymptotic performance should be clearer. Our goal is to obtain strong minimax guarantees in heteroscedastic settings, not to claim asymptotic optimality. We will revise the paper to make this distinction explicit and to discuss more clearly how our method compares conceptually and empirically with algorithms motivated by asymptotic efficiency.
>
> We thank the reviewers again for the helpful feedback. We believe these changes will substantially improve the clarity and strength of the paper.

---

> ### Comment · Reviewer_NUnC · 2026-05-23
>
> Thank you for the response. The revision improves some aspects, especially the added baselines and expanded discussion. However, my main concerns remain. The technical novelty is still not clearly separated from a standard heteroscedastic adaptation of MOSS, and several key proof steps remain too high-level. The experiments are improved but still do not include asymptotically optimal baselines or a convincing minimax-vs-asymptotic comparison (the comparisons are not associated with the corresponding references, which makes it difficult for readers to identify the baselines). There are also several malformed equations in the rebuttal that make parts hard to parse.

---

> > ### Comment · Action_Editor_fTcU · 2026-06-09
> >
> > Dear Reviewer,
> >
> > Please specify which proof steps remain "too high level" so that the authors can fix those. Thank you.
> >
> > Regards,
> > AE

---

> > > ### Comment · Reviewer_NUnC · 2026-06-11
> > >
> > > The proofs are much improved compared to the previous version. However, some issues remain. For example, the manuscript uses both $a^+$ and $a_+$ to denote the same quantity, which may cause confusion and should be standardized throughout the paper.
> > >
> > > I only went through the proof of Theorem 1 in detail. One point that is unclear to me is how the final regret bound becomes $\sqrt{T\sum_{i\ge 1}\sigma_i^2}$. At the end of Step 2, there appears to be an additional term of order $\sigma_1\sqrt{KT}$, and I do not see how this term is absorbed into the final bound.
> > >
> > > More importantly, my main concern is the novelty and significance of the problem itself. In my opinion, the known-variance setting is essentially a straightforward adaptation of the MOSS analysis from the bandit literature. For the unknown-variance setting, closely related results already exist, for example Theorem 4 in *Finite-time Analysis of the Multiarmed Bandit Problem* and the results in *Exploration–Exploitation Tradeoff Using Variance Estimates in Multi-Armed Bandits*. As a result, I do not find the technical contribution sufficiently novel or substantial.
> > >
> > > Therefore, although the paper has been improved significantly in terms of presentation and correctness, my overall recommendation remains **Reject**.

---

### Review · Reviewer_pHr3 · 2026-03-16

**Summary Of Contributions:**

The paper proposes two algorithms to minimize the regret in an MAB problem with heterogeneous reward variances. One assumes that the reward variances are known which can be directly used in the confidence bounds to guide exploration. The other assumes the reward are not known so uses a high-probability variance estimate instead. The authors prove that the first algorithm is minimax-optimal, whereas the second algorithm is minimax-optimal up to a logarithmic factor. The authors provide simple empirical results showing the benefits of variance-awareness in minimising the regret.

**Audience:**

Yes

**Audience Explanation:**

The paper contains simple contributions with relatively limited novelty, and it is written in a clear and accurate way without overclaiming its contributions. However, I'm a bit unsure how interesting this work is to the TMLR audience since most of its results are standard: The lower bound follows standard constructions, and the upper bound amounts to a routine substitution of arm-specific variances into the existing MOSS proof framework. I would appreciate the authors' perspective on what they view as the key technical novelty beyond these adaptations.

The paper would be substantially strengthened by either proving that the logarithmic factor in the unknown-variance upper bound is necessary by adding a new lower bound, or by developing an improved algorithm that closes the gap to the presented lower bound.

**Broader Impact Concerns:**

N.A.

**Claims And Evidence:**

Yes

**Claims Explanation:**

Theorems 1 and 2 prove regret upper bounds which is O(\sqrt{KT}) modulo logarithmic terms. The proof sketches are clear.

**Requested Changes:**

- I'm not sure why upper bounds are labeled as minimax regrets in Theorems 1 and 2. Shouldn't it be lower bounds that are called minimax?
- The figures are hard to parse. Consider using different line styles to denote different types of algorithms.

More minor suggestions:
- use $K$ in maths mode instead of K.
- use ``'' as quotation marks instead of "".
- Avoid using many short paragraphs.
- A naive point: it might be helpful to spell out what "MOSS" stands for the first time it appears.

---

> ### Author Response · Authors · 2026-03-20
> **Response to reviewer**
>
> We thank the reviewer for the careful reading and constructive feedback. We have addressed all of the requested and minor changes in the updated PDF.
>
> Regarding technical novelty, the contribution goes beyond a routine insertion of arm-specific variances into the standard MOSS proof template. The key point is that in the heterogeneous-variance setting, the exploration–estimation tradeoff is fundamentally non-uniform across arms, and our analysis identifies the right variance-dependent quantity that controls the minimax regret. We then design algorithms that exploit this structure in both the known-variance and unknown-variance regimes, and establish matching upper bounds up to logarithmic factors. Technically, this requires new confidence and peeling arguments that account for heteroscedastic rewards while preserving an instance-independent guarantee, which is not immediate from existing homogeneous-variance analyses.
>
> Regarding the comment about the use of the term minimax in Theorems 1 and 2, we agree that the wording could be confusing. Our intent was that these theorems provide *upper bounds that achieve the minimax rate*, while the actual minimax lower-bound statement appears separately in Theorem 3.

---

> > ### Comment · Reviewer_pHr3 · 2026-03-23
> >
> > I thank the authors for their response which has addressed my questions.

---

### Decision · Action_Editor_fTcU · 2026-06-07

**Recommendation:** Reject

**Audience:**

Yes

**Audience Explanation:**

This paper is about multi-armed bandits, which is clearly of interest to the TMLR audience.

**Claims And Evidence:**

No

**Claims Explanation:**

One reviewer NUnC (who is an expert in this line of work) still finds several proof steps too high level. I urge the authors to make everything precise before I recommend the paper for publication.

Please look at the comment by Reviewer NUnc on 11 Jun and revise your paper accordingly.

**Resubmission Of Major Revision:**

The authors may consider submitting a major revision at a later time.